# Experimental Determination of Hydrogen Isotope Transport Parameters in Vanadium

**DOI:** 10.3390/membranes12060579

**Published:** 2022-05-31

**Authors:** Marta Malo, Igor Peñalva, Jon Azkurreta, Belit Garcinuño, Hao-Dong Liu, David Rapisarda, Hai-Shan Zhou, Guang-Nan Luo

**Affiliations:** 1National Fusion Laboratory (CIEMAT), Avenida Complutense 40, 28040 Madrid, Spain; belit.garcinuno@ciemat.es (B.G.); david.rapisarda@ciemat.es (D.R.); 2Department Nuclear Engineering & Fluid Mechanics, Faculty of Engineering, University of the Basque Country (UPV/EHU), Plaza Ingeniero Torres Quevedo, 1, 48013 Bilbao, Spain; igor.penalva@ehu.eus (I.P.); jonazkurreta@gmail.com (J.A.); 3Institute of Plasma Physics, HFIPS, Chinese Academy of Sciences, Hefei 230031, China; liuhd@ipp.ac.cn (H.-D.L.); haishanzhou@ipp.ac.cn (H.-S.Z.); gnluo@ipp.ac.cn (G.-N.L.)

**Keywords:** H isotope transport phenomena, vanadium, permeability, surface rate constants

## Abstract

Deuterium permeation through vanadium membranes in a wide range of pressures and the temperature range ~250–550 °C was experimentally investigated. Measurements on the same material were carried out in three laboratories with different features for an extended characterization and for cross-check validation. A unified equation for deuterium permeability in pure vanadium (99%) was provided as $\Phi =1.27\times 10^{-4}\cdot e^{-8667/T}$ mol m^−1^ s^−1^ Pa^−0.5^, which represents a significant progress for the characterization of the transport properties in this material, given the spread of data, which can currently be found in the literature. Adsorption and recombination rate constants were also measured for hydrogen and deuterium at low pressure for the same range of temperatures. Finally, the influence of the surface roughness was examined by measuring samples with different surface finish.

## 1. Introduction

Hydrogen permeable membranes attract high interest in multiple technological and scientific areas. The search for an alternative to the Pd-based membranes, which involve elevated costs, has revealed vanadium and vanadium-based materials (and other elements from group V) as a promising choice, e.g., for hydrogen separation and purification technologies [1,2,3,4], the main aim being its use as fuel for the production of a carbon-free energy [5,6] or for tritium recycling and exhaust structures in the nuclear fusion field [7,8]. While there is a general agreement on diffusion and solubility values, a wide spread of permeability values is found in the literature [9,10,11]. This might seem illogical, given the direct relationship between the three magnitudes. However, permeability can be considered a complex process in comparison with the more fundamental diffusion and solubility mechanisms, given that it involves multiple steps, including adsorption and desorption, which may be highly influenced by surface conditions.

The traditionally accepted values of hydrogen/deuterium permeability (from now on referred to as “theoretical values”), which were calculated by multiplying solubility and diffusivity data in the 1980s and 1990s [12,13], have not been experimentally validated, and the few attempts to do so have failed over time. Discrepancies have been historically attributed to experimental conditions, such as surface cleaning. Although previous work carried out at the Research Centre for Energy, Environment and Technology, Spain (CIEMAT), suggested a complex dependence on temperature and emphasized the role of the sample surface state [14], a strong recommendation for avoiding the use of the theoretical values was drawn as the main conclusion. The experimental determination and understanding of the H isotope transport parameters in vanadium and other elements of group V are fundamental for the basis of the materials science and must be a priority for the development of hydrogen-related technologies, which in general rely on the high permeability of these metals. Extraction efficiencies miscalculations in the recently proposed prototypes of different laboratories are likely expected [8,15]. In addition, the selection of alternative materials with significantly lower costs might be considered as a function of the results.

Previous results showed remarkable differences with respect to the theoretical values; however, they represented a further contribution to the existent spread of experimental data points. The work underway is focused on clarifying these discrepancies but also aims to provide permeation data for realistic applications. Reliable permeability values in a wide range of driving pressures are provided, with reproducibility being confirmed for the first time by cross-check validation at different laboratories. In addition, the first results for surface parameters have been obtained for this work. Information about the commercial vanadium, which is relevant for permeation characterization (chemical analysis, microstructure, surface roughness), is also provided, which may ease future data validation and further material investigation. Finally, the possible influence of surface polishing was analyzed.

## 2. Materials and Methods

Vanadium samples (99.9% purity) from Eagle Alloys Corporation (Talbott, TN, USA) were analyzed in three laboratories: at the Thermoperm facility belonging to CIEMAT, at PermRIG facility at the University of the Basque Country (UPV/EHU, Leioa, Spain) and at the permeation facility in the Institute of Plasma Physics Chinese Academy of Sciences (ASIPP, Hefei, China). A narrow collaboration of the three institutes was established for the characterization of both the surface (CIEMAT-UPV/EHU) and diffusive (CIEMAT-ASIPP) limited regimes, hence providing a double-check validation of the permeation data.

The chemical analysis of the material showed the following impurity levels (in ppm by weight): Al, 12; Si, 180; Ti + Zr + Hf, <100; P, <53; Mg <50; Nb <50; Sn <50; Ta <50; Fe <20; Cr <20; B <20; W<20; Ni <10; Cu <10; Ca <10; Mo <10; H <10; K <10; Li <10 and Mn <1.

The microstructure of the material is shown in Figure 1, both before and after permeability tests (in Thermoperm). Uniform distribution of predominantly equiaxed grains is found for both as-received and tested samples. The slightly smaller average grain size (about 25 µm and 16 µm for as-received and tested samples, respectively) in the tested samples is likely due to an inherent variation between samples from different batches. Sample preparation, which includes cutting according to the different permeation facilities and surface grinding for thickness reduction and polishing, were carried out at CIEMAT workshops. The roughness of some sample surfaces was measured by using a Bruker DektakXT profilometer (Billerica, MA, USA), the values obtained being <Rq> = 2096.24 nm and 106.74 nm for the as-received and polished samples, respectively.

A summary of the samples analyzed is depicted in Table 1.

The three laboratories make use of the gas-driven permeation (GDP) technique, which consists of injecting a controlled pressure of gas to one of the faces of the testing membrane and measuring the permeation flux at the other side by means of different detection techniques. In the equilibrium, the permeated flux is expressed by Equation (1), so permeability *Φ* is directly obtained:(1)$$J=\frac{D\;K_{s}\;p^{1/2}}{d}=\frac{\Phi \;p^{1/2}}{d}$$
where *D* (m^2^ s^−1^) is the diffusion coefficient, *K*_S_ (mol m^−3^ Pa^−1/2^) is the Sieverts’ constant, *d* (m) is the membrane thickness, *p* (Pa) is the pressure, and *Φ* (mol m^−1^ s^−1^ Pa^−1/2^) is the permeability. The half-power dependence *J*~*p*^1/2^ on the pressure is indicative of pure diffusion-limited regime (DLR). However, under certain conditions (low driving pressures and high temperatures but also favored by thin samples or when a surface contains oxides or contaminants), the transport regime is considered to be surface limited, that is, the limiting process for the permeation of the gas from one side of the membrane to the other is not the diffusion through the bulk but the reactions taking place at the surface. In this case, the flux dependence with the pressure is linear: *J*~*p*. In a pure surface-limited regime (SLR), the adsorption constant, *σ**k*_1_, can be derived from the measured flux as a function of the driving pressure:(2)$$J=\frac{1}{2}\;\sigma k_{1}\;p$$

The recombination constant, *σ**k*_2_, can be derived from the Sieverts’ constant:(3)$$K_{S}=\sqrt{\frac{\sigma k_{1}}{\sigma k_{2}}}$$

Measurements as a function of temperature and pressure are made during the same experimental run either at a fixed pressure with varying temperatures in a number of steps (Thermoperm & PermRIG), or at a fixed temperature by varying the pressure (ASIPP). The different procedures are represented in Figure 2. *Arrhenius* type permeation curves as a function of temperature, and pressure dependence (right insets) are either way obtained, from the stationary phase.

The permeability measurements at CIEMAT were carried out at the THERMOPERM facility [14,16]. The device permits obtaining hydrogen and deuterium permeation data in thin membranes from 300 °C to 550 °C and 0.1 mbar to 1000 mbar. It basically consists of a vacuum chamber, which is divided into two separated and sealed sectors by the membrane to be analyzed. Annealed copper gaskets are used to assure a good isolation. In addition, a vacuum sleeve is connected to a turbo-molecular pump in order to eliminate any possible influence on the measurements of permeated gas through the coupling rings. The testing gas is introduced at room temperature at different pressures in one of the chamber sides, and the permeated gas (through the membrane) is measured at the other side by means of a leak detector (Pfeiffer asm 430, with 10^−12^ mbar minimum measurable leakage for mass 4). Direct heating (10 °C/min) in one of the sample faces is applied by means of a small oven (copper nucleus and mineral insulating Thermocoax with Inconel alloy as heating cable), the temperature being measured with two thermocouples spot welded to the sample. A systematic cleaning procedure of the samples was followed, including previous ultrasonic bath and annealing at 400 °C after the system is completely assembled. For this work, driving pressures between 1 mbar and 1800 mbar were used in order to characterize both SLR and DLR.

The same material was also tested at the Institute of Plasma Physics Chinese Academy of Sciences (ASIPP). The GDP chamber was divided into two parts by the sample: upstream chamber and downstream chamber. Both chambers were first evacuated by a turbo-molecular pump to a background pressure of ~10^−7^ mbar. Then, the sample was heated to the testing temperature by a resistance furnace. A K-type thermocouple was attached to the sample to monitor the testing temperature. The high purity (99.999 wt% purity) D_2_ gas was introduced to the upstream chamber. Gas pressure was precisely measured by a capacitive vacuum gauge (Pfeiffer CMR361, Aßlar, Germany). A quadrupole mass spectrometer (QMS, MKS Microvision Plus, Andover, MA, USA) in the downstream chamber was used for the detection and quantification of the D permeated flux through the membrane, using a D_2_ standard leak (Vacuum Technology Incorporated) for calibration. Masses 3 (HD) and 4 (D_2_) were monitored in order to obtain the permeation flux. Measurements between 20 mbar and 100 mbar were performed for direct comparison with the CIEMAT results (DLR).

At PermRIG facility, with strong expertise in this field [17,18,19], measurements from about 0.3 mbar to 40 mbar and between 250 °C and 550 °C were carried out in order to characterize the SLR. The physical principle of the experimental technique entails the gas flux recording that passes through a thin membrane of the material of interest from a high gas pressure region to a low pressure region at initial vacuum conditions. The hydrogen migration through the specimen is measured by recording the pressure increase with time in the low-pressure region with two capacitance manometers (Baratron MKS Instruments, Andover, MA, USA), P1 and P2, with a full scale range of 10 mbar and 0.13 mbar, respectively. An electrical resistance furnace regulated by a PID controller allows us to establish the sample temperature within a +/− 1 K precision. The temperature of the specimen is measured by a K-type thermocouple inserted into a well drilled in one of the two flanges where the specimen is mounted. The pressure controller allows the instant exposure of the high-pressure face of the specimen to any desired gas driving pressure, which is measured by means of a high-pressure transducer. Before any experimental test is performed with high-purity hydrogen (99.9999%) (or deuterium or any other gas), ultra-high vacuum (UHV) state is reached inside the experimental volumes (below 10^−9^ mbar) in order to assure the absence of any deleterious species (such as oxygen or water vapor) that may provoke surface oxidation of the specimen. Measurements were carried out at the lowest possible pressures in order to optimize the characterization of the SLR.

## 3. Results and Discussion

### 3.1. Permeability

A series of permeability measurements were carried out at Thermoperm (CIEMAT) for a wide range of pressures, from 1 to 1800 mbar. Results for samples #1 and #2 (Table 1) for the permeated flux as a function of the inverse temperature are shown in Figure 3. Results are self-consistent and in good agreement with data reported previously [14], which implies an excellent grade of repeatability (precision).

An identical vanadium sample from the same lot was tested in ASIPP to check the reproducibility of the results. Measurements with deuterium from 400 °C to 500 °C, approximately, and between 20 mbar and 200 mbar were carried out, with similar results being found. Permeability data are shown in Figure 4.

Figure 5 compares the results obtained at 100 mbar at the two laboratories, showing excellent agreement. In a recent paper by T. Fuerst et al. [20] of the Idaho National Laboratory (INL), vanadium permeability was measured in similar conditions of pressure and temperature to those presented in this work but for a different material of similar purity. Thus, Figure 3 is completed with values from Fuerst, showing that the results completely match CIEMAT and ASIPP data.

Figure 5 confirms the results presented in Ref [14], showing a lower permeability (several orders of magnitude) when compared with classical values from Steward (1983) and Reiter (1993) [12,13]. In addition, the trend of permeability with temperature is the opposite to that expected from the theoretical values, having a tremendous impact on the final use and application of this material.

A tentative value for deuterium permeability through vanadium can be extracted from these results (Equation (4)), considering that the measurements carried out at CIEMAT for different samples and multiple experimental runs at pressures between 1 mbar and 1800 mbar give similar values and have been corroborated by both INL and ASIPP institutes.
(4)$$\emptyset_{\left(D_{2}/V\right)}=1.27\times 10^{-4}\cdot e^{-8667/T}\;\mathrm{mol}/(m\;s\;{\mathrm{Pa}}^{0.5})$$

This value was compared in Figure 6 with other potential candidate materials for hydrogen separation systems (tantalum and niobium were not included, given the similar uncertainty about their permeabilities, which also exists [14]), showing the small difference, which exists, for example, with iron, in the range of temperatures considered.

### 3.2. Surface Rate Constants

Surface limited transport regime might apply, mainly depending on the pressure (for instance, the expected tritium partial pressure in fusion components is in the order /below 1 mbar [23], for which SLR is expected) but also on the type of material, the membrane thickness and the temperature. Therefore, the obtained permeability values might not be directly extrapolated to lower pressures and should be measured under exactly these conditions. The surface parameters *σk*_1_ and *σk*_2_, which dominate gas permeation when surface limited regime applies, were determined at UPV/EHU for polished (0.6 mm thickness) and unpolished (1.0 mm thickness) samples, both for hydrogen and deuterium (Table 1). For these measurements, the lowest achievable pressure at each temperature (~0.30 mbar) and temperatures between approximately 250 and 550 °C were selected (Table 2). The *σk*_2_ parameter was calculated from Equation (2), the Sieverts’ constant being taken from the literature [12].

The first results for the adsorption rate constant were also obtained at CIEMAT. Data were calculated from permeation values at the minimum achieved pressure (1 mbar) and for a polished sample, considering SLR. Figure 7 compares data obtained at PermRig with the values from CIEMAT, showing excellent agreement. No related literature has been found on the adsorption constant for pure vanadium.

### 3.3. Surface Polishing

An extensive and systematic study of different parameters, which affect permeability, is underway, including surface finishing. In this work, the first results on the influence of the surface polishing on permeability are shown. The experiments were reproduced under identical conditions for samples with “as received” and polished surface (RMS roughness <Rq> = 2096.24 nm and 106.74 nm, respectively). At CIEMAT, the effect of surface polishing on the permeated flux was evaluated for a DLR, at testing pressure of 100 mbar (Figure 8), and no indication of any influence on the surface finish was found.

However, considerable differences for the polished and unpolished material were found by UPV/EHU in the SLR (Figure 7). A textured surface, i.e., with higher effective surface, would nominally increase the gas permeation flux, since the phenomena taking place at the surface, such as adsorption and desorption, are favored [23]. Even so, different effects might be expected, in particular in materials where resistant native oxides are formed at the surface. The possible contaminants or oxide layers that might be present and that reduce the permeability are more difficult to eliminate on a rough surface. Likewise, surface dislocations and other defects, which can also influence permeability, are modified by polishing. The different behavior obtained at CIEMAT and UPV/EHU for rough and smooth samples can be attributed to the lower range of pressures used at UPV/EHU for the unpolished sample. The effect of surface is higher at lower pressures (pressure is one of the parameters involved in the transition to the surface limited transport regime).

## 4. Conclusions

The latest experimental results on deuterium permeability in vanadium obtained at two different laboratories (CIEMAT and ASIPP) on identical samples confirm that the theoretical values obtained by Steward (1983) and Reiter (1993) should not be applied (at least in ideal situations). Several orders of magnitude lower permeability and opposite behavior with temperature should be expected for effective permeability. A tentative value for deuterium permeability in vanadium for pressures between approximately 1 mbar to 1000 mbar is provided. In addition, the first experimental results for dissociative and recombination parameters, for which no other reference has been found, are given. The current recommendation is the individual experimental analysis of every particular material under the expected operating conditions for a reliable behavior prediction. The role of the surface polishing in permeation was evaluated, results indicating an enhanced permeation for polished surfaces but only at very low pressures, where SLR applies. Future work will include the measurement of solubility by both thermal desorption spectroscopy (TDS) and the absorption–desorption method in order to obtain a complete characterization of the hydrogen isotope transport parameters in vanadium.

## Figures and Tables

**Figure 1 membranes-12-00579-f001:**
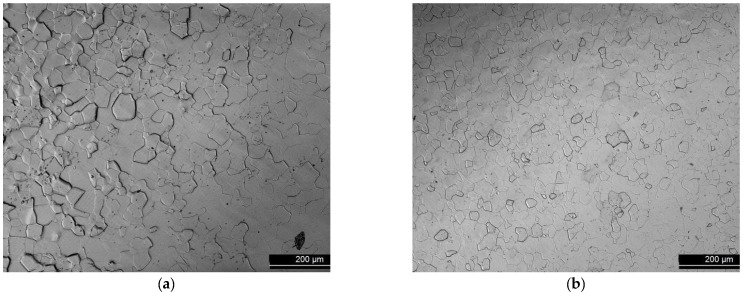
Confocal image for Eagle Alloys material “as received” (**a**), and after being tested at Thermoperm facility with D2 at temperatures ≤500 °C (**b**).

**Figure 2 membranes-12-00579-f002:**
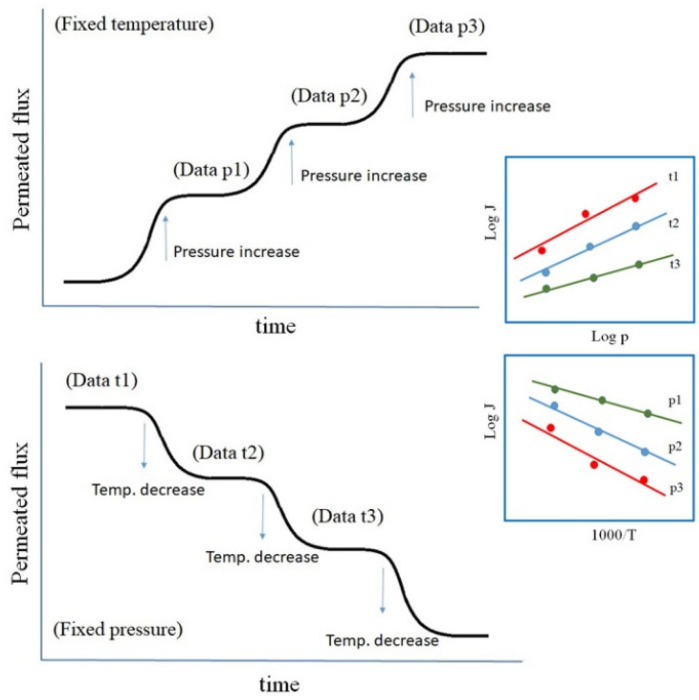
Measuring procedures at ASIPP (**up**), and at Ciemat and PermRiG (**down**).

**Figure 3 membranes-12-00579-f003:**
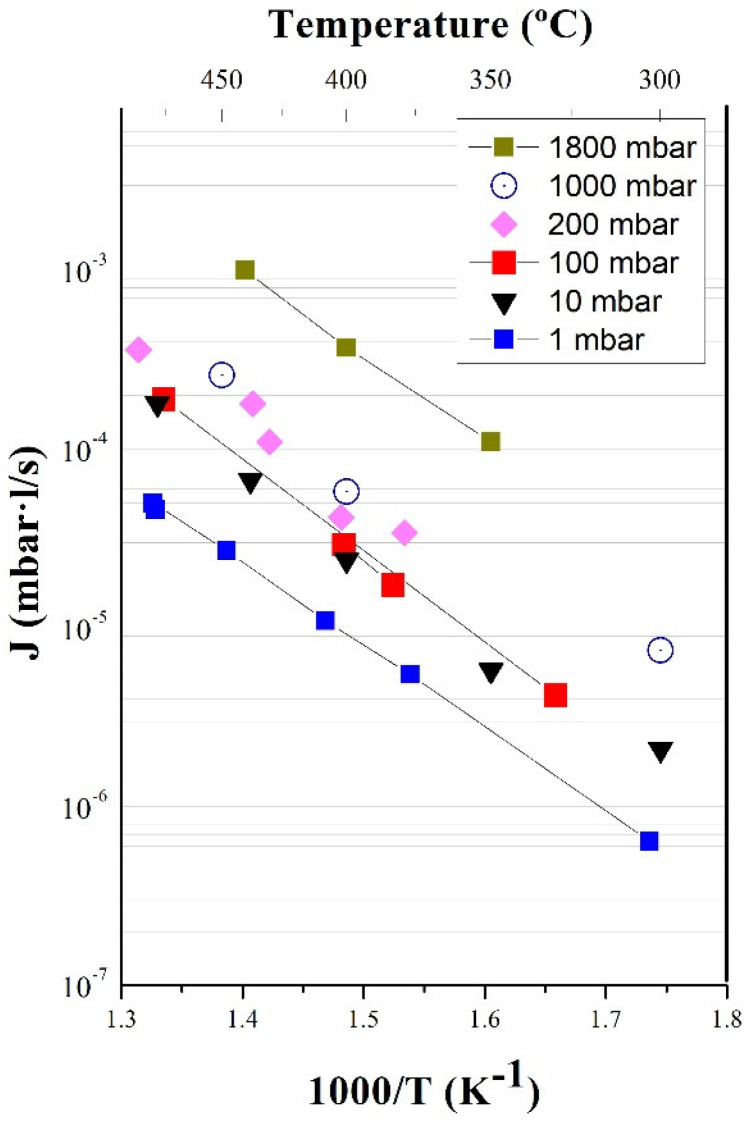
Permeation flux vs. inverse temperature at different pressures (Thermoperm, CIEMAT).

**Figure 4 membranes-12-00579-f004:**
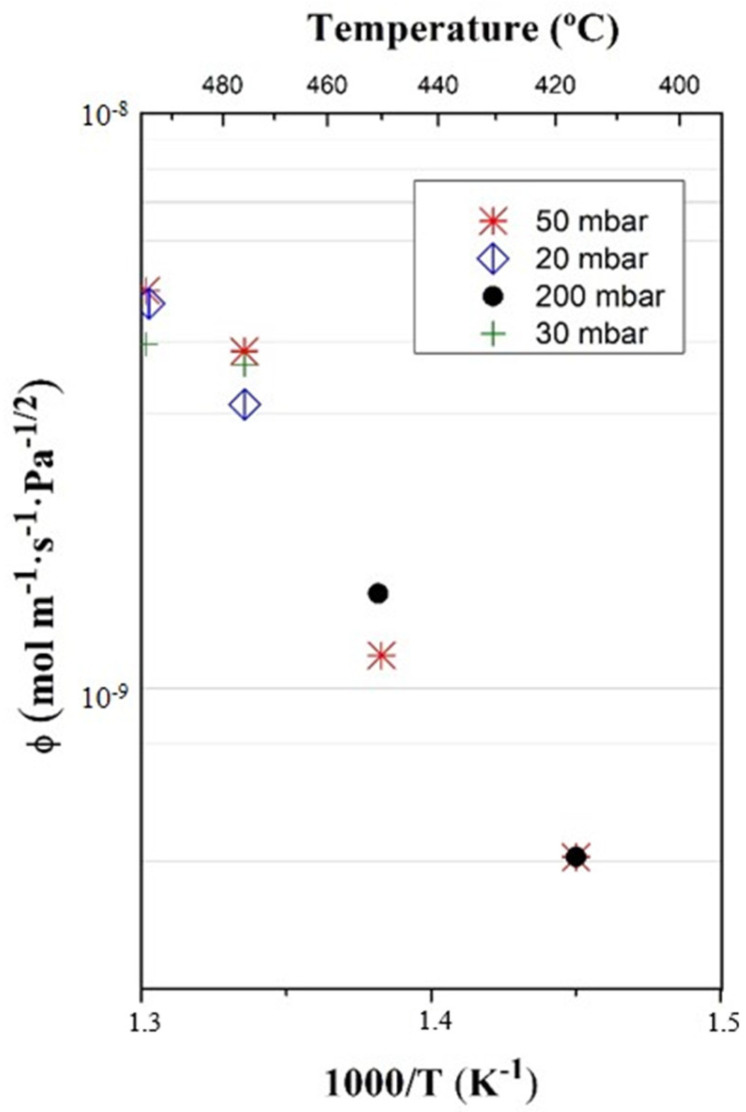
Permeability as a function of the inverse temperature obtained in vanadium (ASIPP facilities).

**Figure 5 membranes-12-00579-f005:**
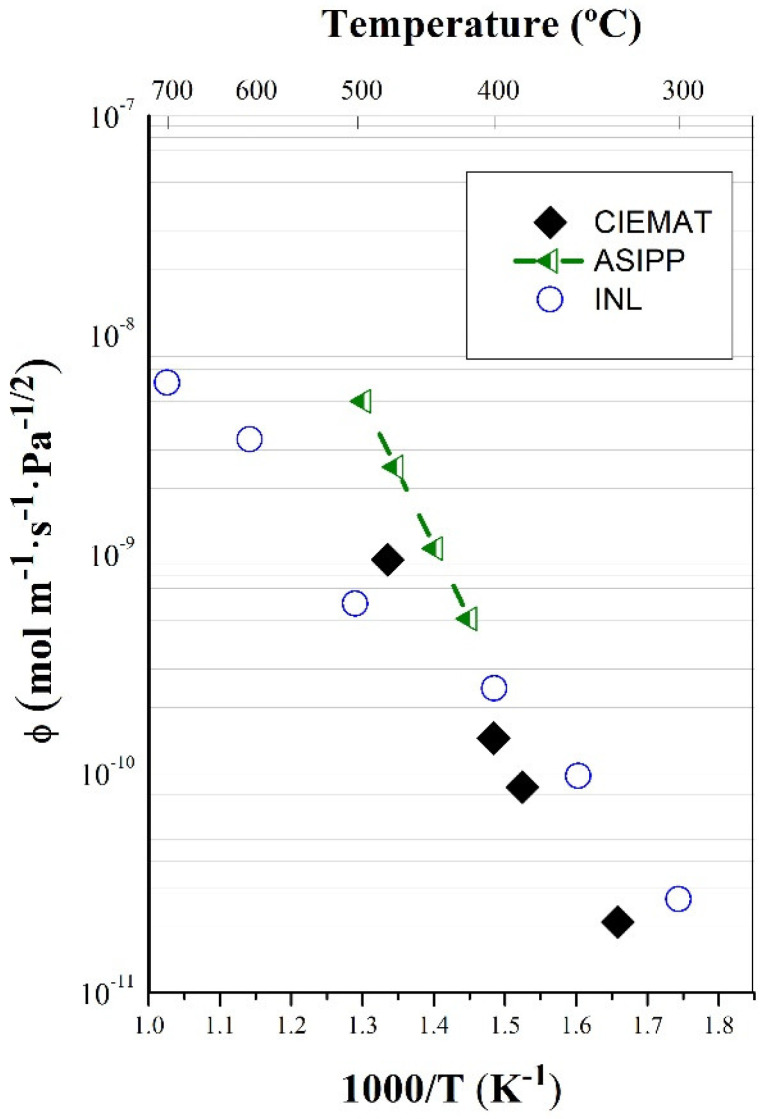
Deuterium permeability in vanadium measured at CIEMAT, ASIPP, and INL [20].

**Figure 6 membranes-12-00579-f006:**
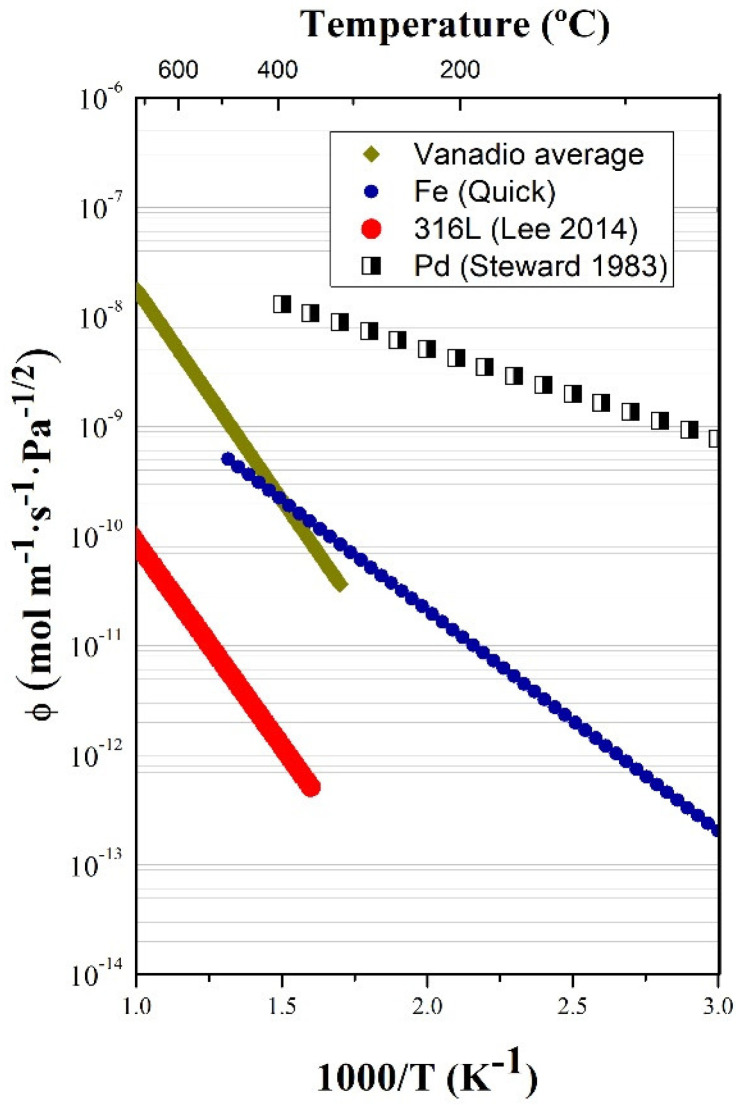
Calculated vanadium permeability (Equation (4)) compared to Fe, 316 L steel and Pd [12,21,22].

**Figure 7 membranes-12-00579-f007:**
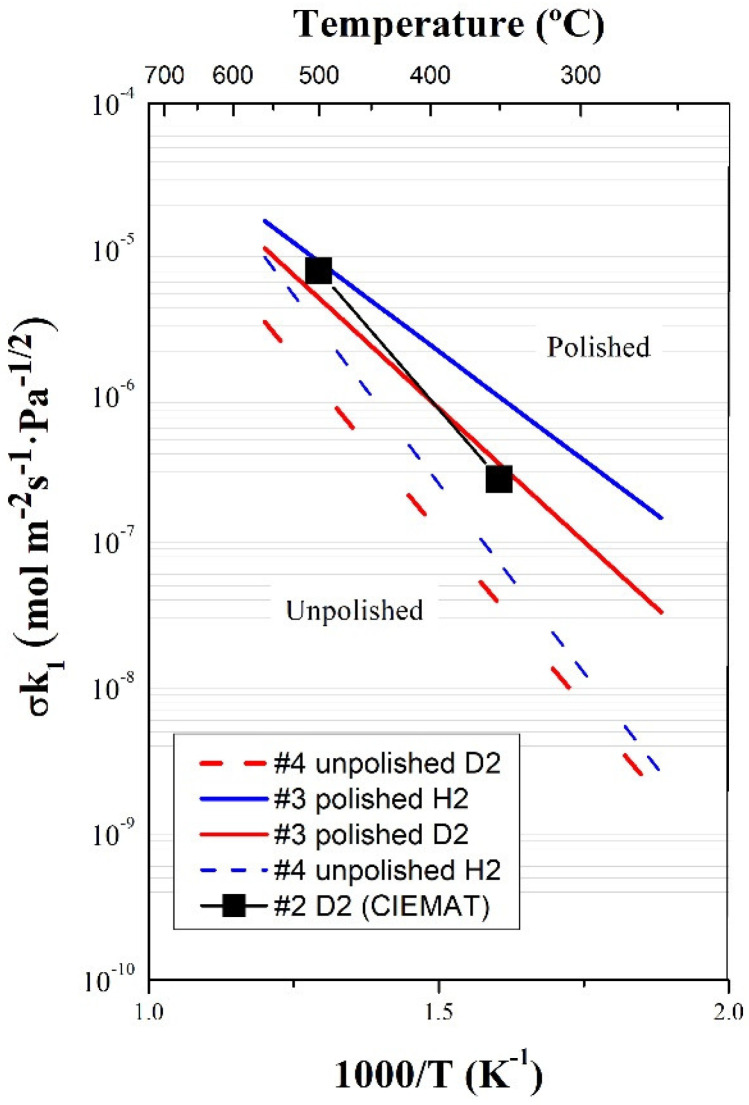
Adsorption rate constant measured at PermRiG and comparison with values from Thermoperm.

**Figure 8 membranes-12-00579-f008:**
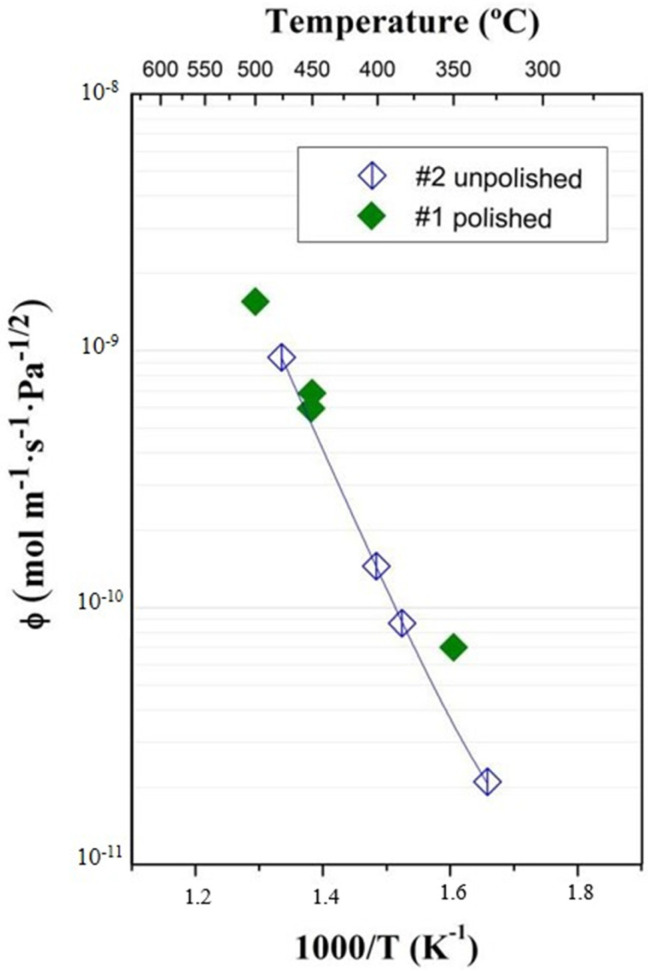
Deuterium permeability through polished and unpolished vanadium samples, 100 mbar.

**Table 1 membranes-12-00579-t001:** Summary of the tests.

Sample	Area mm^2^	Thickness	Surface	*p* (mbar)	Regime	Gas	Institute
#1	38 × 38	0.40 mm	Polished	100–1800	DLR/SLR	D_2_	CIEMAT
#2	38 × 38	1.0 mm	Unpolished	1–100	DLR/SLR	D_2_	CIEMAT
#3	30 × 30	0.60 mm	Polished	0.3–1.8	SLR	D_2_/H_2_	UPV/EHU
#4	30 × 30	1.0 mm	Unpolished	0.3–40.0	SLR	D_2_/H_2_	UPV/EHU
#5	Φ 20 mm	0.96 mm	Polished	20–200	DLR	D_2_	ASIPP

**Table 2 membranes-12-00579-t002:** Surface rate constants.

Sample	Gas	*σk*_1_ (mol/(m^2^ s Pa)) *	*σk*_2_ (mol^−1^·m^4^ s^−1^) *
#3	H	4.72 × 10^−2^*e*^(−6.72*x*)^	2.48*e*^(−1.37×10*x*)^
#3	D	1.92 × 10^−1^*e*^(−8.23*x*)^	1.01 × 10*e*^(−1.52×10*x*)^
#4	H	1.41 × 10*e*^(−1.19×10*x*)^	7.40 × 10^2^*e*^(−1.89×10*x*)^
#4	D	1.73*e*^(−1.10×10*x*)^	9.08 × 10*e*^(+1.82×10*x*)^

* *x* = 1000/*T*.

## Data Availability

The data presented in this study are available on request from the corresponding author.

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
