# Peer review of "Experimental Determination of Hydrogen Isotope Transport Parameters in Vanadium"

_membranes, 2022, doi:10.3390/membranes12060579_

Round 1

Reviewer 1 Report

In a paper entitled "Experimental determination of hydrogen isotope transport parameters in vanadiu", the authors M. Malo, I. Peñalva, J. Azkurreta, B. Garcinuño, H.-D.Liu, D. Rapisarda, H.-S. Zhou and G.-N. Luo investigated the permeation of deuterium through vanadium membranes for different pressures and temperatures in three laboratories by independent groups of researchers to extend characterization and cross-validation. They presented a unified equation for deuterium permeation in pure vanadium. For the same temperature and low pressure range, they determined the absorption and recombination rate constants of hydrogen and deuterium and investigated the effect of surface roughness on their permeation processes. The results of studies on the transport properties of vanadium membranes for hydrogen, presented in this paper, are important for many areas of science and technology, especially for the technology of separation and purification of hydrogen as a fuel for the production of CO2-free energy.

In summary, the authors have solved an interesting issue in the field of membranology. The paper is organized, clearly written and has important cognitive and utilitarian values. However, I noted some shortcomings:

  1. The initial is missing next to Malo (probably M. Malo)
  2. Equation (3) - a mistake or an oversight? Rather a mistake because earlier the authors give the unit KS (mol m-3 Pa-1/2)

Having responded to this suggestion, I recommend the paper for publication.

Author Response

Point 1. The initial is missing next to Malo (probably M. Malo)

Answer1: Yes, it is M. Malo, thank you

Point 2. Equation (3) - a mistake or an oversight? Rather a mistake because earlier the authors give the unit KS (mol m-3 Pa-1/2)

Answer2: σk1 and σk2 have different dimensions: σk1 (mol/(m2 s Pa), and σk2 (mol-1·m4 s-1), hence Sieverts constants dimensions are correct as KS (mol m-3 Pa-1/2)

Reviewer 2 Report

The manuscript describe the experimental deuterium permeability application of vanadium membrane samples obtained from three different manufacturers. The manuscript is easy to read and the research is very crucial towards the development of vanadium based membrane. However, the novelty of the manuscript need to be addressed/highlighted clearly at the end of the introduction.

Author Response

Point 1: The novelty of the manuscript need to be addressed/highlighted clearly at the end of the introduction.

Response: The main novelty of this work is that it presents reliable experimental data which have been validated in different experimental facilities, and in a previously characterized material, in order to ease future comparisons with other works. In addition, it presents the first collection of surface permeation parameters in this material. The last paragraph of the instroduction has been slightly modified so these points are highlighted, as suggested. 

Reviewer 3 Report

Dear Authors,

The strongest point of the manuscript is that authors are reporting results of the study performed in three different laboratories. However, i think that results are not substantiated with enough (raw) data. The possibility of supporting information should be also considered.

In general, having in mind the overall quality of the manuscript, my impression is that thorough enrichment of the text is needed - including experimental base and link to available literature. Thus, I must advice against publication of current submission. 

Kind regards,

Reviewer

Author Response

General comments:

The strongest point of the manuscript is that authors are reporting results of the study performed in three different laboratories. However, i think that results are not substantiated with enough (raw) data. The possibility of supporting information should be also considered. In general, having in mind the overall quality of the manuscript, my impression is that thorough enrichment of the text is needed - including experimental base and link to available literature. Thus, I must advice against publication of current submission.

Answer:

The manuscript has been improved following the reviewer’s suggestions:

A number of experimental runs have been carried out at both CIEMAT and ASIPP laboratories for an extensive range of pressures which confirm both repeatability and reproducibility of the results for permeability data. However, only average data was presented in the case of ASIPP. A new figure showing extra data points obtained at ASIPP has been now included. This provides a better visualization of the main strengths of this work, which is the reproducibility of the measurements in three different laboratories.

New references were included in the introduction for a better understanding of the issue (spread of the experimental data points in the literature) although a broad description can be found in a previous reference (ref 14)

A detailed description of the experimental techniques and several references are provided, including an extra figure to clarify the measuring procedure. In addition, minor spell and punctuation marks corrections have been made for a general improvement of the text.

Minor spell and punctuation marks corrections have been made, according to the reviewer's comments.